# Waste Polyurethane Foams as Biomass Carriers in the Treatment Process of Domestic Sewage with Increased Ammonium Nitrogen Content

**DOI:** 10.3390/ma16020619

**Published:** 2023-01-09

**Authors:** Ewa Dacewicz, Anna Lenart-Boroń

**Affiliations:** 1Department of Sanitary Engineering and Water Management, Faculty of Environmental Engineering and Land Surveying, University of Agriculture in Kraków, Adam Mickiewicz Ave. 24/28, 30-059 Kraków, Poland; 2Department of Microbiology and Biomonitoring, Faculty of Agriculture and Economics, University of Agriculture in Kraków, Adam Mickiewicz Ave. 24/28, 30-059 Kraków, Poland

**Keywords:** biomass carrier, polyurethane foams, domestic wastewater, microbial community

## Abstract

In order to understand the mechanisms of microbial growth on waste polyurethane sponge materials, their effectiveness as biomass carriers in domestic sewage with increased ammonium nitrogen content treatment was assessed. Comparative experiments were carried out in microreactors under steady conditions of batch culture, which allowed for an assessment of different carriers, in the form of flexible foams, rigid foams, and flexible foams placed in full casings. In the studies conducted in continuous cultures, biomass carriers selected in batch culture were used as fillings in the column model. The structure of the microbial community inhabiting the spongy material was determined and the pollutant-removing process from real domestic sewage was assessed. Analyzes using the Illumina sequencing technique allowed for demonstrating that *Nitrosomonas* and *Nitrospira* were the predominant nitrifiers in the biomass carrier in the form of waste polyurethane foams (PUF). It was found that anammox bacteria, the presence of which—as unidentified *Planctomycetes*—was confirmed in the polyurethane sponge material, were also responsible for the high removal of N-NH_4_^+^. *Burkholderia* and *Sphingopyxis* phyla were identified as the dominant denitrifying bacteria involved in the treatment of domestic sewage with increased content of ammonium nitrogen. The biomass carrier in the form of waste PUF placed additionally in full casings proved to be more beneficial for the proliferation of bacteria involved in nitrification and denitrification processes. On the other hand, waste foams without casings proved to be more suitable for the growth of microorganisms known to perform partial denitrification and may accumulate nitrites (*Staphylococcus*, *Dokdonella*). Additionally, the presence of *Devosia* and *Pseudonocardia*, which participated in the phosphorus removal process, was found in the waste PUR foams.

## 1. Introduction

One of the methods of intensifying wastewater treatment processes is the immobilization of microorganisms on properly selected carriers. It aims to limit the movement of microbial cells by immobilizing them, which results in increased availability of nutrients. The immobilization process takes place with the use of biomass carriers, which cannot adversely affect the viability of cells, their division, and growth. A properly selected carrier should effectively retain immobilized microbial cells, be mechanically and chemically stable, and have sufficient porosity to allow free diffusion of products and substrates. Such material should also be able to be used on a technical scale, be easily available and its production cost should be low [1,2].

An interesting example of biomass carriers is spongy materials, of which polyurethane foams (PUF) have gained importance recently. Due to the high porosity, and thus the specific surface of the material, open-pored polyurethane foams deserve a lot of attention [3,4]. Among the special properties of PUFs is their hydrophobic nature, which enables them to interact with most microbial cell surfaces [5,6]. The hydrophobicity of foams decreases when they are covered with biomass [7]. Increasing the water absorption capacity makes the foams resistant to drying out. Microorganisms thrive best in a humid environment, therefore the ability of the material to absorb water is an important factor in selecting the carrier, especially with its periodic sprinkling.

In the literature, much attention is paid to a simple method of PUF regeneration by rinsing adsorbed pollutants with distilled water [8], as well as their low investment cost and mechanical strength. In recent years, polyurethane foams have been studied as carriers for immobilized bacterial cells to remove crude oil from seawater [9], decolorize or bioremediate industrial wastewater [10,11], treat dairy wastewater [12], remove toxic pollutants from industrial wastewater [4], as well as for municipal wastewater treatment [2,13].

Polyurethanes (PUR) are synthetic materials, that have been widely used both in industry and in everyday life. They may appear, among others, in the form of foam materials, divided into rigid foams (used as insulation and packaging material) and flexible foams (used for the production of upholstered furniture, mattresses, sleeping pillows, and car seats). Currently, polyurethanes rank sixth in the world production of polymers. According to forecasts, their production will increase from 54.0 to 75.3 billion USD in 2021–2026 [14,15,16].

During the production of flexible polyurethane foams, industrial waste, the so-called scrap is also produced. The second type of PUF waste is post-consumer waste resulting from material usage. Most polyurethane waste is incinerated or placed in a landfill. Both of these methods of waste management are not indifferent to the environment. The first one results in the emission of toxic and greenhouse gases. When depositing PUF waste in a landfill, it should be remembered that it is not biodegradable. In addition, the low density of polyurethane foams makes storage uneconomical due to the requirement of a large area of land. For the above reasons, innovative solutions for the management of waste PUR foams should be sought. One of the ways is to use them as a material for the production of new products (material recycling) or as a substitute for the original raw material (raw material recycling). In 2016, post-consumer waste in the US accounted for 110 thousand tons of recycled polyurethanes, and its ratio was only 5.5% [14]. An example of the management of flexible PUF scrap waste is its use as a filling for pillows or soft toys. Dacewicz et al. [17,18,19] proposed the use of post-consumer waste in the form of PUR foam scraps as a filling of vertical flow filtration columns in the treatment process of domestic sewage with increased content of ammonium nitrogen.

This article describes research aimed at understanding the growth mechanisms of microorganisms on polyurethane foams, by assessing their effectiveness as biomass carriers. The carrier pre-selection, which was completed in microreactors, was based on the procedure reported by Bolton et al. [20]. It included an evaluation of the efficiency of eight waste materials, determined by the relative biological activity of bacterial cells immobilized on each carrier. The experiment was carried out in batch cultures under steady environmental conditions. This allowed for a comparative assessment of the biological activity of microbial cells between carriers. In order to assess the scalability of this procedure, subsequent experiments were carried out in continuous cultures, using selected biomass carriers as filling in the column model. The structure of the microbial community inhabiting the spongy material was determined and the process of removing inorganic forms of nitrogen from real domestic sewage with increased content of ammonium nitrogen was assessed.

## 2. Materials and Methods

### 2.1. Research Materials

#### 2.1.1. Biological Material

Distilled water was used as a blank medium, which was inoculated with 1% (*v/v*) inoculum. Real wastewater was used as the experimental medium and inoculated with 1% (*v/v*) inoculum.

*Inoculum* was a commercial bio-preparation containing effective microorganisms (EMs) in the form of two different populations of microorganisms, i.e., yeast and lactic acid bacteria. The total number of *Saccharomyces cerevisiae* yeast in this preparation was 5.0 × 10^3^ CFU/cm^3^, while the total number of lactic acid bacteria *Lactobacillus casei* and *Lactobacillus* were 5.0 × 10^6^ CFU/cm^3^.

The EM-containing culture was added in liquid form into 100 cm^3^ of the medium.

#### 2.1.2. Selection of the Type of Biomass Carriers

The selection of PUR foams as biomass carriers was based on (1) material porosity, (2) its affinity for the growth of microorganisms, and (3) its tolerance for long-term use. An equally important factor was (4) the dimensional stability of the biomass carriers [7,8]. A characteristic feature of spongy materials is their water absorption. Water-soaked PUR foams increase in weight, which causes their deformation. The dimensional stability of biomass carriers can be ensured by their greater rigidity or additional casing, which makes them more resistant to compaction.

Two types of biomass carriers were selected for preliminary research. The first type includes post-consumer waste material of polyurethane foams in the form of their scrap. Porous soft (flexible) foams and closed-pore rigid (inelastic) foams were used (Figure 1). Soft foams (bluish-green Figure 1a, yellow-green Figure 1b, orange Figure 1c, lilac Figure 1d) were characterized by open pores with different contents. Soft foams with large cells in the structure were less flexible and had lower mechanical strength than foams with finer pores. Due to the denser and more compact cell structure, the latter is characterized by high flexibility and thus is more resistant to mechanical damage. The selection of porous foams was made based on the diameter and size distribution of their cells, porosity, specific surface area, and application properties, which are documented in detail in the previous papers of the authors [8]. The closed-pore rigid PUFs used in the study are shown in Figure 1e,f. This material came from the post-consumer waste of thermal insulation. Inelastic green PUFs were characterized by high strength and greater dimensional stability compared to white polyurethane foams. The irregular shapes of both rigid foams had average dimensions of 3 × 30 mm (width and length).

The second type of carrier was a combination of waste materials: in the form of soft PUR foams and pieces of conduit (Figure 1g). The conduit was used as a full casing of the soft foam in order to ensure its additional dimensional stability. The pieces of the conduit used in the tests came from a gray corrugated flexible polyvinyl chloride (PVC) pipe, cut into 2.0–2.5 cm long pieces with a diameter of 16 mm/11 mm (external and internal dimensions).

The above biomass carriers were divided into three groups (Figure 1): porous flexible foams (group I), closed-pore rigid foams (group II), and porous flexible foams placed in casings (group III).

#### 2.1.3. Batch Culture of Microorganisms

Batch cultures were carried out in the experimental and control medium under the conditions of full mixing with identical parameters (temperature of 25 °C, agitator rotation 200 rpm). Biomass carriers from groups I–III were added to the experimental medium. The control medium was the experimental medium without the addition of a carrier. Thorough mixing of the culture was aimed at removing the substrate without being limited by mass transport.

### 2.2. Methods

#### 2.2.1. Microbial Growth Rate Determination

Microbial growth rate determination in biomass batch cultures was used to select the proper carrier from group I. This stage of research was based on the determination of the optical density of the cell suspension. Batch cultures of microorganisms were conducted for 54 h in triplicates. Small laboratory flasks with a capacity of 250 cm^3^ were used as microreactors in this study. A weight of flexible PUFs (group I) of about 0.6 g was introduced to each reactor, and 50 cm^3^ of real sewage after the septic tank, 50 cm^3^ of distilled water, and 1 cm^3^ of inoculum were dosed. The control sample was a sample without the addition of the carrier (100 cm^3^ of sewage + 1 cm^3^ inoculum).

At 0, 1, 2, 3, 4, 5, 6, 7, 8, 20, 24, 28, 32, 48, and 54 h of the batch culture, samples were taken and used as material for further analysis. It involved the determination of the actual number of cells in the culture by measuring the optical density (OD). After the end of the culture (after 54 h) the dry biomass content was also determined.

#### 2.2.2. Selection of the Type of Biomass Carriers

The biomass carriers for continuous culture were selected from groups I-III. The selection was based on determining the optical density of the produced cell suspension and the content of biomass immobilized on carriers. Group I included two types of flexible polyurethane foams selected in the previous stage: bluish-green (Figure 1a) and yellow-green (Figure 1b). Microreactors with a volume of 250 cm^3^ were used in this study. The weight of the carrier of 0.6 g (foam) and/or 1.8 g (conduit) was introduced into each microreactor, along with 100 cm^3^ of sewage after the septic tank and 1 cm^3^ of *inoculum*. The average value of COD and the average concentration of N-NH_4_^+^ in raw sewage were 246 mgO_2_·dm^−3^ and 170.3 mg·dm^−3^, respectively. The C/N ratio was 1.44.

Batch cultures of microorganisms in this part of the study were carried out for 54 h in triplicate. At 0, 2, 4, 6, 8, 18, 20, 24, 28, 32, 48, and 54 h of the batch culture, samples were collected and used as the material for further studies. These included the determination of the actual number of cells in culture by measuring optical density (OD). After the end of the cultivation (54 h), the dry biomass content was also determined.

#### 2.2.3. Ammonium Nitrogen Removal in Continuous Cultures

The effect of increasing the scale on the effectiveness of the selected material as a biomass carrier was assessed by conducting experiments on a laboratory scale. A mixture of waste soft foams was used as the filling material, in which the bluish-green foams constituted approx. 30%, yellow-green foams approx. 20%, lilac foams approx. 10% and orange foams approx. 33% [8]. Before the start, the sponge material was inoculated with active sludge from the MBBR reactor and fed with sewage with increased content of ammonium nitrogen.

The two-column models of the vertical flow filter were built on a semi-technical scale and operated continuously for 14 months without additional aeration. The same amount of actual domestic sewage pre-treated in the septic tank was fed to each column. A detailed description of the test stand and a detailed composition of raw wastewater have been included in the previous papers [17,18,19,21].

This research compared the effectiveness of the nitrogen removal process during the treatment of wastewater with increased content of ammonium nitrogen in two columns. The first one was filled with a mixture of waste flexible foams placed additionally in casings (column model A), while the second one was with waste soft foams without casings (column model B).

In order to determine the structure of biomass carriers after they were colonized by microorganisms, SEM microscopic analyzes of the sponge material from columns A and B were performed. Microscopic observations and the structural analysis of the microbial community that inhabited the polyurethane foams used as the biomass carrier in both columns were also performed. The microbial community composition that inhabited both types of biomass carriers during the treatment of domestic sewage in continuous cultures was analyzed based on the 16S rRNA gene sequences.

### 2.3. Determinations

#### 2.3.1. Optical Density Determination

Prior to the experiments, the waste PUR foams were rinsed with sterilized distilled water in order to remove any undesirable contaminants from their surfaces. The next step was the sonication process, which consisted in immersing each carrier in 0.9% NaCl. The last step in determining the optical density was the measurement of the absorbance of biomass in the tubes with the Spectroquant spectrophotometer at a wavelength of 560 nm (OD_560_). A total of 0.9% NaCl solution was used as a blank.

#### 2.3.2. Biomass Content in the Tested Carriers

The calculation of dry biomass consisted in determining the weight of the carrier before and after the cultivation. For this purpose, weights of carriers were dried to constant weight at 70–100 °C and then weighed on a Radwag Ws 220/C/2 analytical balance.

#### 2.3.3. Microscopic Observations of Biomass

Sediment samples taken from representative PUFs were mixed before microscopic observation. A 0.5 cm^3^ sample of sediment was placed on a microscopic slide in order to determine the presence of higher organisms, i.e., ciliates, rotifers, flagellates, and nematodes. The microscopic observations were carried out using a Bresser Science MPO 401 (Bresser GmbH, Rhede, Germany) preparation microscope.

#### 2.3.4. Scanning Electron Microscope (SEM) Observations

Scanning electron microscope photos (Jeol, JSM- 5500 LV; JEOL Ltd., Tokyo, Japan/Lanameter MP-3 microscope) were used to determine the morphology of the biomass inhabiting the analyzed foams. Prior to observations of 3D structures, air-dry samples of the spongy material were cut into 1.5 cm thick elements with a microtome blade, then they were gold sprayed with the Jeol JFC 1200 ion coater. High-resolution photos were taken with a cooled charge-coupled camera (Photometrics model CH 250 charge-coupled device, Tucson, Ariz.).

#### 2.3.5. Illumina Sequencing of 16S rRNA Gene

Sponge samples (approximately 10 mg dry weight) were collected from the bottom layer of the spongy material from the A and B column models. Bacterial genomic DNA was extracted using Genomic Mini AX Bacteria + (A & A Biotechnology, Gdańsk, Poland), followed by DNA purification using Anty-Inhibitor Kit (A & A Biotechnology, Gdańsk, Poland). DNA concentration was measured in a Qubit 4 Fluorometer. The presence of bacterial DNA in the examined samples was confirmed by Real-Time PCR in an Mx3000P (Stratagene, La Jolla, CA, USA) thermal cycler using SYBR Green as a fluorochrome and Universal 16S rRNA primers [22].

The amplicon libraries of V3-V4 regions within the 16S rRNA gene were prepared according to the 16S Metagenomic Sequencing Library Preparation Part # 15044223 Rev. B (Illumina, San Diego, CA, USA), followed by a two-step PCR using Herculase II Fusion DNA Polymerase Nextera XT Index Kit V2. The library quality was verified according to Illumina qPCR Quantification Protocol Guide. The sample libraries were loaded on Illumina MiSeq Platform and 2 × 300 bp reads were generated by Macrogen (Seoul, Korea).

Briefly, Illumina Next Generation Sequencing technology is also called high-throughput DNA sequencing. It allows for delivering data output ranging from c.a. 300 kb to more than tb of nucleotide sequence reads in a single instrument run. It simultaneously provides data on, e.g., the entire bacterial population composition within a single sample. It comprises the following steps: DNA sequencing libraries generation by clonal amplification by PCR. Then, the DNA is sequenced by synthesis, i.e., the DNA sequence is determined by the addition of nucleotides to the complementary strand. Next, the spatially segregated DNA templates are sequenced simultaneously [23,24].

#### 2.3.6. 16S rRNA Gene Sequence Analysis

The 16S rRNA V3-V4 regions from the Illumina sequencing were identified by examining the sequence reads against the Greengenes v.13 database (97% similarity, minimum score 40). The resulting sequences were taxonomically assigned at the phylum level or lower ranks. CLC Genomic Workbench v.12 (Qiagen, Venlo, The Netherlands) and Microbial Genomics Module Plugin v.4.1. (Qiagen, Venlo, The Netherlands) were used to measure alpha diversity indices, such as the Shannon-Wiener diversity index, Simpson index, and Margalef’s richness index. These are three measures used for the characterization of population diversity. The Shannon-Wiener diversity index takes into account the number of species living in a habitat (richness) and their relative abundance (evenness). The higher the Shannon-Wiener index value, the higher the population diversity. The Simpson index takes into account the number of species present, as well as the abundance of each species. The lower the index value, the higher the population diversity. Finally, Margalef’s richness is a count of the number of different species in a given area or community [25].

#### 2.3.7. The Efficiency of the Nitrogen Removal Process

The efficiency of the ammonium nitrogen removal process was assessed on the basis of the degree of its conversion into organic forms: nitrite nitrogen (first phase of nitrification) and nitrate nitrogen (second phase of nitrification). The removal of nitrate nitrogen (denitrification process) was also taken into account. Determinations of non-ionic forms of nitrogen were performed in filtered samples. In order to determine the concentration of ammonium, nitrite, and nitrate nitrogen, colorimetric methods were used with the WTW Photolab S12 spectrophotometer (WTW GmbH, Weilheim, Germany).

### 2.4. Statistical Analyzes

Optical densities were compared by analysis of variance using the Statistica v.13 program. The statistical significance between the control medium and the experimental media (flexible foams, rigid foams, flexible foams placed in full casings) was determined using the non-parametric Kruskal–Wallis test for a significance level of α = 0.05.

## 3. Results and Discussion

### 3.1. Microbial Growth Rate Determination

Figure 2 shows the kinetic curves of microbial culture growth in fully mixed batch microreactors. The determined curves clearly show the stage of microbial growth and its decline. For the control medium, the optical density and thus the growth rate were the lowest. Within 8 h of cultivation using flexible PUR foams as cell carriers, an increase in OD is clearly visible, which reflects the growth of the microbial cultures. Taking into account the growth dynamics, microorganisms multiplied most slowly in the presence of orange PUFs, as evidenced by the smallest angle of the curve in this phase. The lowest OD value was found in the culture with orange foam with the largest Feret diameter of 1.53 mm. In the cultures with the remaining foams, the OD value was higher by approx. 30%. This means that the conditions provided by the presence of bluish-green and yellow-green PUFs with a Feret diameter in the range of 0.50–0.67 mm were more favorable to the process of microbial multiplication in flexible open-pored foams.

There was a significant difference between the OD of the control medium and the experimental media (Kruskal-Wallis test, *p* < 0.04). No significant differences in optical density were observed between the individual flexible foams (Kruskal-Wallis test, *p* > 0.05).

Table 1 presents the mean values of dry biomass immobilized on carriers in the form of flexible PUR foams after 54 h of batch culture with full mixing.

Yellow-green and bluish-green foams, characterized by similar porosity and the lowest degree of hydrophobicity [8], showed a clearly higher mean amount of absorbed biomass (approx. 0.2 g/1 g of weighed sample). Three times less biomass was absorbed on the surface of the orange and lilac foam. This was due to the lower percentage of pores of orange (61.3%) and lilac (53%) foams compared to other foams (approx. 63%) [26]. In the case of the least porous lilac foams, only a few micropores can be observed on their smooth skeleton surface, while the skeleton of the remaining foams was covered with numerous microporous structures [8]. According to research on DHS reactors, the open porous structures of the sponges facilitated water diffusion and gas volatilization, and their hydrophilicity ensured adequate contact between the substrates and the population of microorganisms [7]. The smallest amount of immobilized biomass, i.e., 0.06 g/1 g of the sample, was found on the carrier in the form of an orange foam, characterized by the highest degree of hydrophobicity. For the above reasons, flexible yellow-green and bluish-green foams were selected for further research on the selection of the carrier, divided into groups I-III.

### 3.2. Selection of the Type of Biomass Carriers

Figure 3 shows the optical density values for groups I–III. The lines marked for carriers in the form of flexible foams (bluish-green and yellow-green) clearly show the stage of microbial growth and their death. For the bluish-green foams in casings, the growth phase was more dynamic (greater inclination angle) and the stationary phase occurred between 10 and 20 h of culture. The biomass cell death phase started after 20 h of culture. In the case of rigid foams with closed pores (white and green) and the conduit itself, there was no linear phase of microbial biomass growth, only cell death.

The highest affinity to microbial growth was observed during 48 h of culture in the presence of bluish-green elastic foam with open pores, additionally placed in a full casing. Similar values of optical density occurred during culture in the presence of bluish-green and yellow-green PUFs. The correlation coefficients between the OD values and the materials used to indicate similar growth conditions for microorganisms for both open-pore flexible foams (correlation coefficient 0.88), both closed-pore rigid foams (correlation coefficient 0.94), and the flexible bluish-green foam without casing and the one additionally placed in a conduit (correlation coefficient 0.82).

Table 2 presents the biomass content of microorganisms adsorbed on the tested carriers. The highest amount of biomass, i.e., 0.7 g/1 g of the sample was found on the filling which was a combination of pieces of conduit with elastic bluish-green foam with open pores. This value was more than three times higher than in the case of bluish-green foam without additional casing. Among the carriers from group I, the biomass content was observed at the level of 0.2 g/1 g of the sample. Group II carriers, i.e., rigid foams, characterized by closed pores, adsorbed three times fewer microorganisms. The lowest number of microorganisms was found in group III on the pieces of conduit without additional filling.

Group I flexible foams with open pores were selected for the next stage of research on a laboratory scale. In order to differentiate the biomass carriers, apart from the bluish-green and yellow-green foams of similar porosity, more porous (orange) and less porous (lilac) foams were used for the tests. The results of research by Uemura et al. [27], who emphasized that the spongy material with smaller pores was characterized by better COD and ammonium nitrogen removal through better oxygen uptake, were also taken into consideration.

### 3.3. Ammonium Nitrogen Removal in Continuous Cultures

Detailed results of studies on the removal of organic pollutants from real domestic sewage with an increased ammonium nitrogen content have been included in previous works by Dacewicz [17,26,28]. The author reports, that there was high average efficiency of removing easily biodegradable organic substances marked as BOD_5_ (c.a. 85%) and hardly biodegradable substances marked as COD_Cr_ (c.a. 80%) in the column filled with a mixture of waste foams placed additionally in casings [26]. In the case of the column filled with foams without casings, the average BOD_5_ and COD_Cr_ removal efficiency were lower by 15 and 10%, respectively.

Due to the low C/N ratio observed in the treated domestic sewage, aerobic heterotrophic bacteria did not displace nitrifying bacteria in both columns. Dacewicz [19] stated that in the upper spongy layer, the activity of heterotrophic bacteria was much higher than that of autotrophic AOB bacteria. On the other hand, significant differences in the removal of non-ionic forms of nitrogen were observed in the spongy layers of both columns. This fact prompted the Authors to carry out a microscopic analysis and an analysis of the microbial community composition in the waste spongy material, which was the filling of columns A and B.

### 3.4. Microscopic Observations of Biomass

Low production of biomass, caused by predation of macrofauna, is a characteristic element of reactors filled with spongy material [29,30]. The presence of higher organisms, i.e., sedentary ciliates (*Vorticella* sp.), rotifers (*Philodina* sp.), and nematodes (*Nematoda n.det.*), which resulted in increased bacterial removal as a result of predation (Appendix A) was demonstrated in columns A and B. This fact is pointed out by many researchers studying the UASB-DHS system [29,31,32]. *Cladocera* and *Oligochaeta*, whose presence was reported by Onodera et al. [29] were not observed in our study.

### 3.5. Scanning Electron Microscope (SEM) Observations

Figure 4 shows SEM pictures of bluish-green, yellow-green, orange, and lilac foams taken at a magnification of 5000×. Fluffy biomass of microorganisms with similar morphology was observed in the bluish-green and yellow-green foam matrices in both columns (Figure 4a–b). The presence of a few single colonies of cocci and rods attached to the biomass (Figure 4a–c) or directly to the polyurethane foam structure (Figure 4d) was also observed. Ribeiro et al. [33], when analyzing the SEM photos, showed that the immobilization of biomass in the brand-new PUR foams was based on the formation of microgranules, which were mechanically retained in the porous structure of the carrier or on its surface. These authors also found the presence of diffuse cells attached directly to the PUR foams. Varesche et al. [5] in their research pointed out that both fluffy biomass and individual microorganisms were firmly attached to the surface of PUR foam using van der Waals forces and hydrophobic bonds. In the subject material of waste PUR foams, we found the presence of bacterial cells attached to the carrier in the form of dispersed, microgranules or compact biomass. The SEM photos show that the sediment accumulated on the surfaces and inside the PUFs is denser for foams with smaller pores. The high amount of biomass in spongy support materials can affect the efficiency of the nitrogen removal process. Chu and Wang [34] found that higher nitrification rates in the PUF carrier were due to the greater amount of biomass. On the other hand, the accumulation of biomass in spongy material with small pores may cause its blockage [35] and, as a consequence, even clogging [8,19,26].

The SEM photos (Figure 5) show the most porous orange foams and the least porous lilac foams showing a fluffy form of biomass and single bacterial colonies attached to the inner surface of the polyurethane. The structure of the most hydrophobic orange foam (Figure 5a) showed significant roughness which helped to bind microorganisms to its surface. In the case of the lilac foam (Figure 5b), its flat surface is visible at 5000× magnification. Although it is not conducive to the fixation of the biomass on its surface, the open pores of the PUFs helped to trap water and biofilm. Similar observations regarding the surface roughness of materials were reported by Dorado et al. [36] in their research on the use of PUFs in the biofiltration process.

### 3.6. Microbial Community Analysis

The microbial community structure of the two types of PUR foams was analyzed based on the 16S rRNA gene sequences. A total of 118,320 reads (51,056 in sample A—PUR foam in the conduit and 67,264 in sample B—sole PUR foam without conduit) were analyzed. Using a 97% sequence identity cut-off, 828 OTUs were detected in sample A and 872 in sample B (Table 3). The diversity indexes, i.e., Shannon-Wiener, Simpson, and Margalef’s, indicate high, but at the same time similar level of biodiversity of the examined samples [25]. In general, it is assumed that bacterial population diversity is a positive phenomenon, as it contributes to the stability and resilience of microbial communities when they are challenged by their surroundings [37]. The mechanisms driving the positive relationship between taxon diversity and population resilience may be related to the fact that more diverse communities are more likely to contain taxa with complementary response traits and the ability for rapid compensatory growth after disturbance [37]. This also might be the case for microbial communities inhabiting biomass carriers used for the sewage treatment process, because sewage composition is rarely constant and the probability that the dwelling conditions for bacteria become difficult, is high. Among the detected OTUs, the members of *Proteobacteria* phylum predominated in both samples (i.e., 74.76 and 67.81% in samples A and B, respectively). The second and third most frequent phyla were: in sample A: *Actinobacteria* (7.99%) and *Planctomycetes* (3.80%); in sample B: *Actinobacteria* (6.32%) and *Acidobacteria* (6.08%). The dominance of *Proteobacteria* has been also observed by other authors in their studies on sponge reactors [38,39,40]. The prevalence of most phyla was similar in both samples, only in the case of *Acidobacteria*, *Chloroflexi*, and *Cyanobacteria* the differences in the relative abundance among the two samples were more distinct, in favor of sample B, where the % reads of these groups were approx. twice higher (Figure 6A,B).

The phyla *Proteobacteria*, *Chloroflexi*, and *Planctomycetes* have been often reported in anammox reactors [41,42,43], which may indicate that this type of reaction also occurred in the examined samples. Anammox bacteria are able to oxidize ammonium into dinitrogen gas under anoxic conditions [44]. Therefore, even though none of the seven to-date described in the literature genera capable of anammox metabolism have been found in our study, the possibility of this process occurring in both samples cannot be ruled out. According to Pereira et al. [45], *Proteobacteria*, *Chloroflexi*, and *Planctomycetes* phyla have always been found in anammox reactors. Additionally, *Chlorobi*, *Acidobacteria*, and *Bacteroidetes* are often found in these systems. All these groups were found in this study and their total share amounts to 86.53% in sample A and 86.42% in sample B.

In terms of the differences at the OTU level, microbial community compositions differed clearly between the two examined samples (Figure 7).

Bacteria belonging to the order *Rhizobiales* were the most prevalent in both samples, but their abundance in sample A significantly outnumbered sample B (36.76% vs. 17.14%). One of the possible reasons for such a situation might be the lower availability of free oxygen in sample A (PUR foam in the protective pipe) which would promote the proliferation of bacteria capable of respiration under microaerobic conditions, such as rhizobia [46]. Members of *Rhizobiaceae* have been studied for their nitrification and aerobic denitrification capabilities and they might play important role in sewage treatment processes. Members of the family *Rhizobiaceae* and the order *Rhizobiales* were found to be among the top three relatively abundant bacterial populations in experimental sewage treatment ecosystem samples, with relative abundance ranging from 4.9% to even 60.5% [47]. The second most frequent group in our study belonged to the family *Xanthomonadaceae*, but these bacteria were much more prevalent in sample B (12.43%), whereas in sample A their share was 4.70% and it was smaller than that of genus *Burkholderia* (5.88%) and family *Rhodospirillaceae* (5.35%). Members of *Xanthomonadales* were also found by Yang et al. [47] in the experimental sewage treatment ecosystem in relative abundances ranging from 10.5 to even 78%. These are bacteria capable of performing heterotrophic denitrification, similar to the members of *Burkholderiales* [41], which were also found by Yang et al. [47] among the top three most abundant populations in one of the samples derived from experimental sewage treatment system with a relative abundance of 9.9%. This group includes *Burkholderia* sp., which was on the other hand the most abundant genus in sample A and prevailed over their abundance in sample B (24.11% vs. 8.69%; Figure 8). The genus *Aquicella*, which was, on the other hand, the most prevalent in sample B, was also found among the dominant microorganisms in a tidal-flow constructed wetland, aimed at wastewater treatment and proved to have nitrogen removal properties [48]. Figure 8 shows the most prevalent bacterial genera identified in the examined samples. The majority of these bacteria are capable of denitrification, nitrification, and other—not yet precisely determined—nitrogen removal properties. Therefore, similarly as concluded by Mahajran et al. [38], the abundant presence of these microbial communities (Figure 8 and Figure 9a,b) indicates the simultaneous occurrence of nitrification-denitrification processes occurring in the examined reactors [49]. However, what needs to be mentioned, is that the abundance of nitrifying bacteria in the examined samples is much lower than that of denitrifying bacteria (Figure 9a,b), which has been also reported by other authors in microbial community analysis in sponge reactors [40,50]. In columns A and B, *Nitrosomonas* constituted 0.34 and 0.76% of all reads, respectively, while *Nitrospira* constituted 0.65 and 0.48%, respectively (Figure 9a).

Phosphate-removing bacteria were also identified in our study (Figure 9d). Both columns showed a similar number of reads for *Pseudonocardia* (*Actinomycetes*), which is capable of excess phosphate accumulation [51]. The presence of *Proteobacteria* belonging to the genera *Devosia*, *Acinetobacter*, and *Bdellovibrio* was also observed. In the bottom spongy filling without casings (column B) there were 2.6 times more reads for *Devosia.* Zuo et al. [52] showed that *Devosia* sp. and *Bdellovibrio* sp. predominated in the process of oxygen capture of phosphorus, while *Acinetobacter* sp. played a dominant role in its anaerobic release. In this study, the above-mentioned phosphorus-removing microorganisms accounted for 1.03 and 1.00% of all reads in columns A and B, respectively.

### 3.7. Efficiency of Ammonium Nitrogen Removal Process

Figure 10 shows the concentration of N-NH_4_^+^, free ammonia (FA), nitrite accumulation rate (NAR), and the ammonium nitrogen removal rate (ARR) in the two spongy layers of columns A and B.

The tests showed that in the outflow from the upper layer of column A, the oxygen content was on average 3.94 mg·dm^−3^, and the mean pH was 7.56. The filling in the form of waste foams placed in the casings allowed for obtaining an average of 37.3% efficiency of nitrification in this layer (Figure 10a). Therefore, there were suitable conditions not only for the development of bacteria responsible for the removal of organic carbon [8,19,26] but also for AOB bacteria, as indicated by the NAR amounting to an average of 67.4%. According to Anthonisen et al. [53] the FA limit, which inhibits the growth of NOB (*Nitrite Oxidizing Bacteria*), ranges from 0.1 to 1.0 mg·dm^−3^. In the outflow from the upper layer of column A, the mean content of free ammonia FA was 1.79 mg·dm^−3^, contributing to the inhibition of bacterial growth of phase II nitrification.

In the upper layer of column B, the efficiency of the nitrification process was lower. The mean oxygen concentration in the outflow was also lower and amounted to 3.85 mg·dm^−3^. The mean pH value in the outflow was higher, i.e., 7.72. The filling in the form of waste foams without casings allowed for obtaining the nitrification efficiency in this layer at an average level of 19.9%, which proved to be almost half lower compared to column A (Figure 10b). In the outflow from the upper spongy layer of column B (Figure 10b) nitrite accumulation was higher (80% on average). In the case of the nitration stage, the inhibition of NOB bacterial activity was influenced by the concentration of free ammonia in the upper layer of waste foams of 3.16 mg·dm^−3^, which appeared to be higher than in the case of column A.

The wastewater flowing into the lower spongy layer of column A was characterized by an average concentration of ammonium, nitrite, and nitrate nitrogen of 99.0 mg·dm^−3^, 10.0 mg·dm^−3^, and 4.4 mg·dm^−3^, respectively. The mean oxygen concentration and pH in the outflow were 4.02 mg·dm^−3^ and 5.24, respectively. In this layer, the dominant phyla contributed to the circulation of not only carbon [8,19,26] but also nitrogen. Dacewicz [19] reports that in the middle part of column A, with a C/N ratio of 1.3, the nitration stage carried out by the NOB bacteria had an advantage over the ammonium nitrogen oxidation stage. Filling in the form of waste foams placed in casings was more favorable for the development of bacteria of the II nitrification phase because the average concentration of free ammonia at the level of 0.18 mg·dm^−3^ did not inhibit their growth. Under these conditions, the ammonium nitrogen removal rate (ARR) was at a similar level as in the upper layer (38.1%), and the nitrite accumulation rate (NAR) decreased to 52.6%. This indicates that the 1st nitrification phase was less efficient in the lower spongy layer of column A. The amount of *Nitrosomonas* in this layer proved to be three times lower compared to column B (Figure 9a), which was affected by the lower concentration of inflowing ammonium nitrogen. According to Dacewicz the value of Y_NO2/NH4_ ratio exceeded the value of Y_NO3/NH4_ three times, which indicates the occurrence of another form of NH_4_^+^ removal. Illumina sequencing revealed the presence of *Planctomycetes*, which have always been found in anammox reactors. As shown in Table 3, the core microbiome of nitrifying and anammox biomass in waste foams additionally placed in casings was formed by *Proteobacteria* (74.76% of sequences), *Actinobacteria* (7.99%), *Planctomycetes* (3.80%), *Acidobacteria* (2.83%), *Bacteroidetes* (2.07%), and *Chloroflexi* (1.85%), which in total accounted for 93.30% of the microbial community. Wang et al. [54] and Yang et al. [55] in their research noted that the growth of phylum *Proteobacteria* was associated with high efficiency of NH_4_^+^-N removal from synthetic sewage with an ammonium nitrogen content of 50 and 150 mg·dm^−3^, respectively. Bulgarelli et al. [56] and Wang et al. [57] found on the other hand a significant contribution of *Actinobacteria* to nitrogen transformation in wetlands. In our study, *Proteobacteria* and *Actinobacteria* phyla prevailed in column A as compared to column B (Table 3).

Sewage flowing into the lower spongy layer of column B was characterized by the mean concentration of ammonium, nitrite, and nitrate nitrogen of 126.1 mg·dm^−3^, 9.8 mg·dm^−3^, and 2.5 mg·dm^−3^, respectively. The mean oxygen concentration and pH value in the outflow from this layer were 4.87 mg·dm^−3^ and 7.30, respectively. In this layer of column B, the efficiency of the nitrification process was at a lower level compared to column A. Despite the fact that the number of *Nitrosomonas* in this layer was three times higher (Figure 9a), the ARR was lower and averaged 32.9%.

In the lower spongy layers of columns A and B, the efficiency of phase II of nitrification was similar. Analysis of the microbial community composition in the PUR waste foams showed a similar amount of *Nitrospira* in both carriers (Figure 9a). As reported by Kubota et al. [58], *Nitrosomonas* and *Nitrospira* were identified as nitrifying bacteria in a DHS reactor treating domestic sewage with an ammonium nitrogen content of 30 mg·dm^−3^. In a column filled with foams placed in casings, the number of reads of *Nitrosomonas* was almost two times lower compared to *Nitrospira*. Such a relationship was also observed by Watari et al. [59] during the removal of ammonium nitrogen of 100 mg·dm^−3^ in a single-stage mainstream anammox process using a sponge-bed trickling filter.

In column B, *Nitrosomonas* appeared to be the dominant nitrifier compared to nitrite-oxidizing *Nitrospira*. Free ammonia, remaining on the mean level of 0.98 mg·dm^−3^, limited the growth of NOB bacteria and inhibited the nitration stage. However, the reduction in the degree of nitrite accumulation by 22% to 57.9% suggested that their removal took place not only through nitrification. In the lower spongy layer of column B, the conditions for the growth of anammox bacteria were optimal according to Strous et al. [60], i.e., pH in the range of 6.7–8.3, and the NH_4_^+^/NO_2_^−^ ratio of 2.5 was closer to the stoichiometric value of 1/1.32. Illumina sequencing showed that in column B the number of *Planctomycetes* responsible for the anammox process was 33% higher compared to column A.

As shown in Table 3, the core microbiome of nitrifying and anammox biomass in waste foams without casings consisted of *Proteobacteria* (67.81% of reads), *Actinobacteria* (6.32%), *Acidobacteria* (6.08%), *Planctomycetes* (5.06%), *Bacteroidetes* (2.94%), and *Chloroflexi* (3.07%) which in total accounted for 91.28% of a microbial community. Zhao et al. [61] showed that *Chloroflexi* can oxidize nitrite nitrogen. The importance of filamentous bacteria of the phylum *Chloroflexi* group in the formation of anammox granules is also known. As reported by Martins et al. [62], *Chloroflexi* appears to be associated with their formation and structure maintenance. A high abundance of *Chloroflexi* in biofilms was also reported by Chen et al. [43] in the long-term treatment process of 50 mg/L of ammonia wastewater in a one-stage partial nitritation and anammox with the bio-carriers system. In our study, there were more *Chloroflexi* phylum representatives in column B compared to column A (Table 3). The microscopic pictures taken at a magnification of 5000 (Figure 4 and Figure 5) show that the fluffy biomass of microorganisms is more frequent than microgranules.

Taking into consideration bacteria responsible for both the nitrification and denitrification process (*Planctomycetes* and *Cyanobacteria* phyla), their number in column B was greater than in column A. As reported by Kubota et al. [58], *Cyanobacteria* phylum was identified in sponge reactors inoculated with activated sludge.

Analysis of the microbial community composition revealed that denitrifying genera *Sphingopyxis*, *Rhodoplanes*, and *Hyphomicrobium* were present in both carriers (Figure 9c). The latter two genera were also observed by Watari et al. [39] in studies on the treatment of wastewater from natural rubber processing using the DHS reactor. Bacteria of the genus *Burkholderia* were identified in the 2.6-times higher amount in foams placed in casings. The greater number of these bacteria could have been influenced by the greater amount of nitrate nitrogen substrates flowing to the lower spongy layer of column of A. Wang et al. (2020) found that when the NO_2_^−^/NH_4_^+^ ratio is in the range of 0.1–0.5, denitrifying bacteria and anammox bacteria can coexist. In our research, the NO_2_^−^/NH_4_^+^ ratio fluctuated within these limits and in the case of columns A and B, it was on average 0.35 and 0.29, which according to Wang et al. [63] suggests that the rate of denitrification contribution was much higher than that of anammox bacteria.

Removal of inorganic forms of nitrogen in the process of autotrophic nitrification and anaerobic denitrification requires the provision of aerobic and anaerobic conditions, respectively, as well as an additional carbon source in the denitrification process. In recent years, research has been conducted on the removal of nitrogen from wastewater by heterotrophic nitrification/aerobic denitrification (HNAD) bacteria, which include *Pseudomonas*, *Zooglea*, *Dechloromonas*, *Flavobacterium* phyla [64,65]. In the presence of a sufficient amount of organic carbon and oxygen, HNAD bacteria are able to convert ammonium nitrogen NH_4_^+^-N and its nitrification forms (NH_2_OH, NO_2_^−^N, and NO_3_^−^N) into N_2_ or N_2_O in the processes of simultaneous nitrification and denitrification [66]. Research on the partial denitrification-anaerobic ammonium oxidation process for municipal wastewater treatment is also worth noting. Among the denitrifiers that accumulate nitrites during denitrification, *Staphylococcus* sp. is mentioned [65]. Si et al. [67] reported that *Zoogloea*, *Dechloromonas*, and *Dokdonella* are microorganisms known to perform partial denitrification. Kirishima et al. [68] reported that the genus *Dechloromonas* was the most abundant in the bottom part of the DHS reactor during the treatment of low-strength municipality sewage in the UASB-DHS system. Nomoto et al. [69] reported that *Zoogloea* and *Dechloromonas* were not found in significant numbers in down-flow hanging sponge reactors. In our research, the above-mentioned HNAD bacteria were also not detected in a significant amount as they constituted only 0.18 and 0.31% of all reads. The share of *Staphylococcus* and *Dokdonella* detected in sample B was greater (0.32 and 0.18% of all reads) than in sample A (0.21 and 0.08% of all reads, respectively).

The occurrence of types of bacteria removing non-ionic forms of nitrogen differed in both samples. *Planctomyces*, which can remove ammonium nitrogen by the anammox process, accounted for 1.31 and 1.90% of all reads in samples A and B, respectively (Figure 11).

In the case of nitrifiers and bacteria carrying out both nitrification and denitrification, their amount in sample A was 25% and 12% higher, respectively, than in the case of sample B. There were also clear differences between the two samples in the relative abundance of bacteria carrying out full denitrification. Twice as many of these bacteria were found in sample A, while 68% more bacteria responsible for partial denitrification were found in sample B (Figure 11A,B). A process called partial denitrification-anammox (PD-AMX), which occurred to a greater extent in PUF waste without casings, has been of increasing interest in recent years, and also there has been increasing interest in mainstream municipal wastewater treatment [65].

A high degree of inorganic nitrogen removal of 51.5% was observed in the two spongy layers of column A filled with waste foams in full casings. Sewage flowing out of the lower layer of column A was characterized by the mean concentration of ammonium, nitrite, and nitrate nitrogen of 60.9 mg·dm^−3^, 8.3 mg·dm^−3^, and 7.1 mg·dm^−3^, respectively. In column B, where the PUR foams were without casings, the efficiency of removing non-ionic nitrogen forms was lower by 15.6%. This was affected by smaller numbers of denitrifiers (Figure 11B). The mean concentration of ammonium, nitrite, and nitrate nitrogen in wastewater discharged from the lower layer of column B was 84.8 mg·dm^−3^, 9.7 mg·dm^−3^, and 6.4 mg·dm^−3^, respectively. Apparently, the high removal of N-NH_4_^+^ was caused by nitrifying bacteria and anammox bacteria, the presence of which was confirmed in the spongy filling of both biomass carriers.

## 4. Possible Future Prospects Associated with the Use of Waste PUF as Biomass Carriers

The use of waste PUF as biomass carriers in a system consisting of a septic tank and a vertical flow filter proved to be an efficient way for direct treatment of domestic sewage. Compared to the traditional biological nitrogen removal technology, which includes the processes of autotrophic nitrification and anaerobic denitrification (e.g., in an SBR reactor), the proposed solution is less complicated. It does not require strict control of aerobic and anaerobic conditions and an additional carbon source in the denitrification process. The use of high-performance spongy materials also ensures a smaller total filling volume and thus the compact size of the device. This economical—due to the lack of the need for external aeration—technology, compared to most of the existing oxygen systems, is an interesting solution for the biological treatment of domestic sewage with increased content of ammonium nitrogen.

## 5. Summary and Conclusions

Next-generation sequencing-based research on the community composition of microorganisms inhabiting waste PUR foams revealed a large variety of microorganisms observed in the spongy material. Both columns show the presence of *Planctomycetales*, which play a significant role not only in the carbon cycle but also in the nitrogen cycle. Many genera and species within this phylum are capable of anaerobic anammox oxidation of ammonium nitrogen. Illumina sequencing showed that *Planctomycetes* phylum accounted for 3.80 and 5.06% of all reads in samples A and B, respectively.

The dominant taxa of denitrifying bacteria were the representatives of Betaproteobacteria (*Burkholderia*) and Alphaproteobacteria (*Sphingopyxis*). In the column filled with waste foams in full casings, the denitrification process was carried out mainly by *Burkholderia*, and their number was 2.6-times greater than in the spongy material without casings. Our research also identified microbial populations of the genus *Devosia*, *Acinetobacter*, *Bdellovibrio*, and *Pseudonocardia*, that were involved in the removal of phosphates.

It was demonstrated that the ammonium nitrogen removal from domestic sewage with its increased content took place in all layers of spongy material. On the one hand, the stiffening in the form of a conduit improved the compaction resistance of flexible foams, and on the other hand, it provided the most favorable conditions for the development of ammonium-removing bacteria. The filling in the form of waste foams placed additionally in the casings was more favorable for the development of the phase II nitrification bacteria because the concentration of free ammonia did not limit their growth. The efficiency of N-NH_4_^+^ removal proved to be the highest and it reached more than 60% in two layers of this waste material. It was found that a short process of nitrification and denitrification could take place in the column filled with flexible foams without additional stiffening in the form of full casings. The prevailing conditions inhibited the growth of NOB bacteria (partial nitrification at FA > 1 mg·dm^−3^) and allowed the growth of anammox bacteria.

The conducted research shows that waste polyurethane foams in the form of scraps of upholstery sponges can be used as an effective biomass carrier in the treatment of domestic sewage with increased content of ammonium nitrogen. The use of additional stiffening in the form of a conduit on the one hand minimized the thickening of the foams, and on the other—ensured high efficiency of the nitrogen removal process. The process of nitrification and denitrification can be successfully carried out in PUF waste placed additionally in full casings. In the case of combining the process of nitrification and partial denitrification-anammox (PD-AMX), it is proposed to use PUF waste without casings.

Further research should be performed to study the long-term using waste PUF as biomass carriers in industrial-scale applications in wastewater treatment.

## Figures and Tables

**Figure 1 materials-16-00619-f001:**
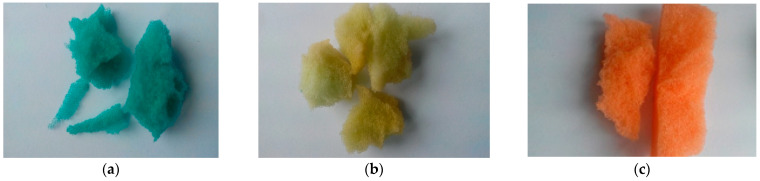
Biomass carriers used in the study, (**a**–**d**) group I, i.e., porous flexible foams, (**e**,**f**) group II, i.e., rigid foams with closed pores, (**g**) group III, i.e., porous elastic foams placed in a conduit.

**Figure 2 materials-16-00619-f002:**
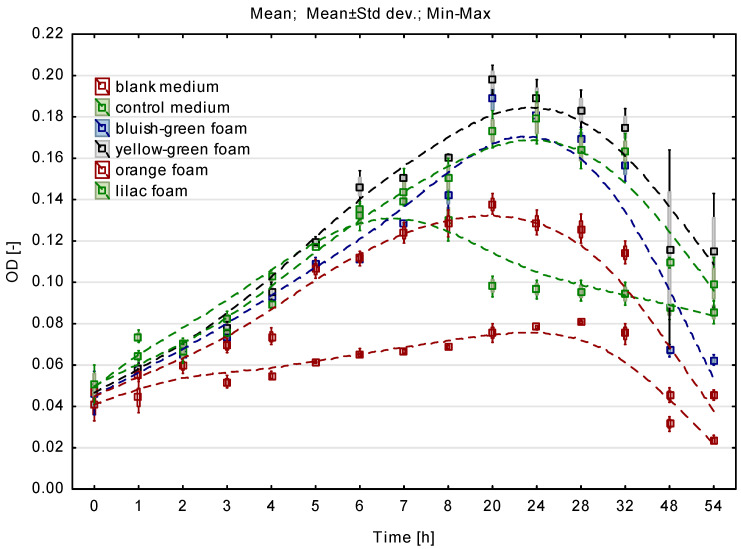
Optical density for batch culture using open-pore flexible PUFs (microbial culture growth kinetic curves).

**Figure 3 materials-16-00619-f003:**
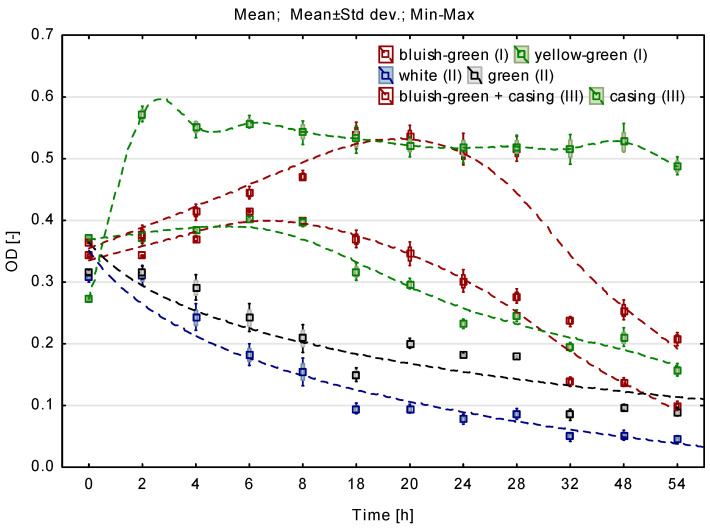
Optical density for batch culture using flexible PUFs from groups I–III.

**Figure 4 materials-16-00619-f004:**
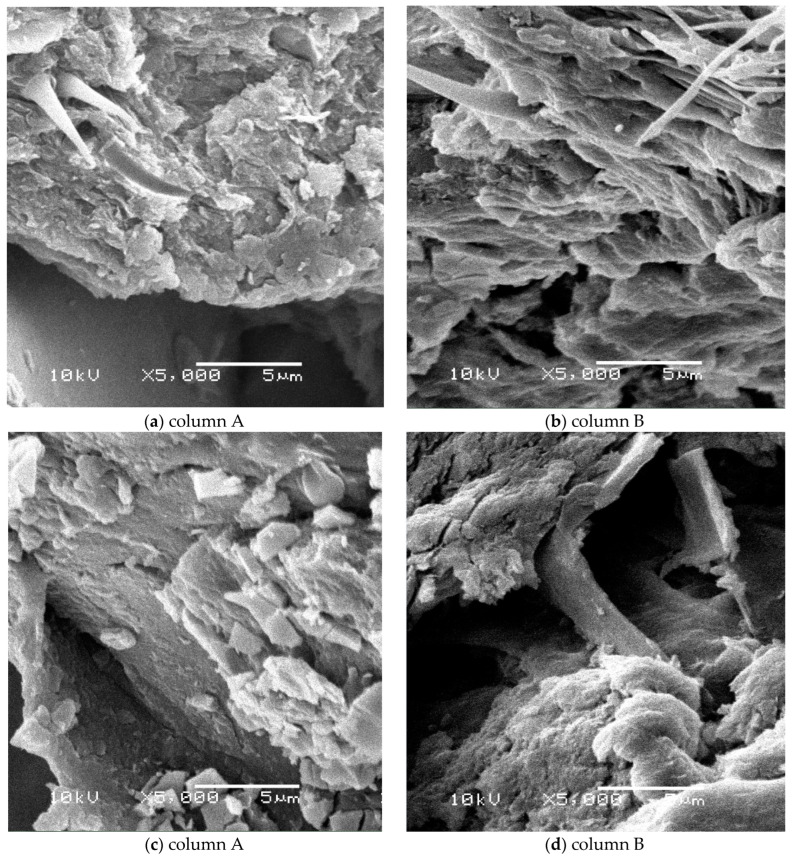
Microorganisms observed in the structure of waste polyurethane foams (**a**,**b**) bluish-green; (**c**,**d**) yellow-green (single colonies of bacteria are marked with arrows).

**Figure 5 materials-16-00619-f005:**
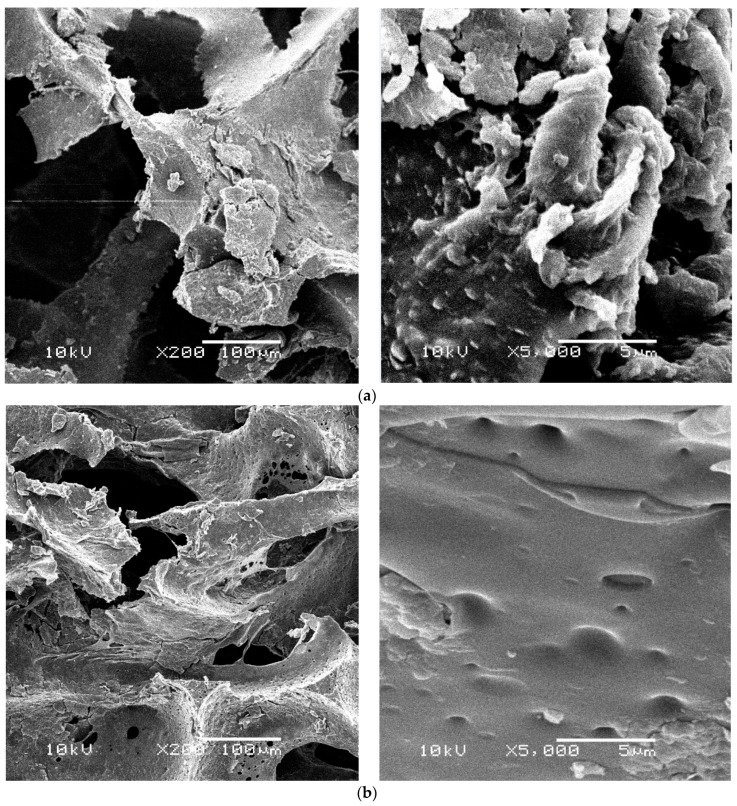
The SEM images for a cross-section of PUFs, (**a**) orange; (**b**) lilac (arrows indicate fluffy biomass fb; single colonies of bacteria sc).

**Figure 6 materials-16-00619-f006:**
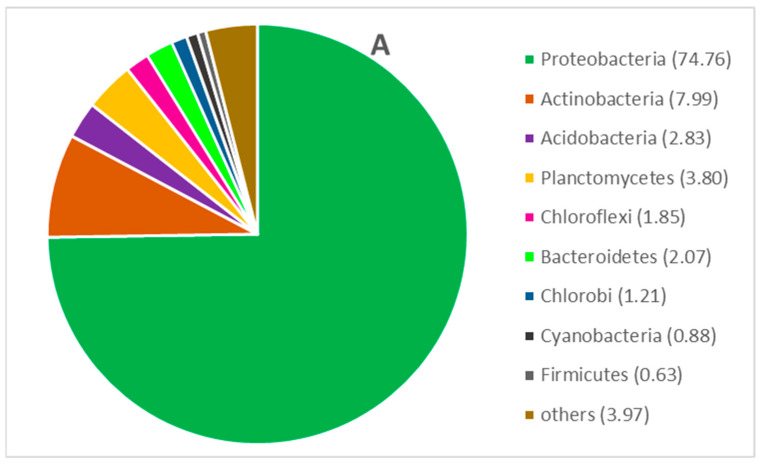
Bacterial community structure shown as the percentage share of different phyla in the two types of PUR foam samples (figures (**A**) and (**B**) refer to the PUR foam samples (**A**) and (**B**), respectively).

**Figure 7 materials-16-00619-f007:**
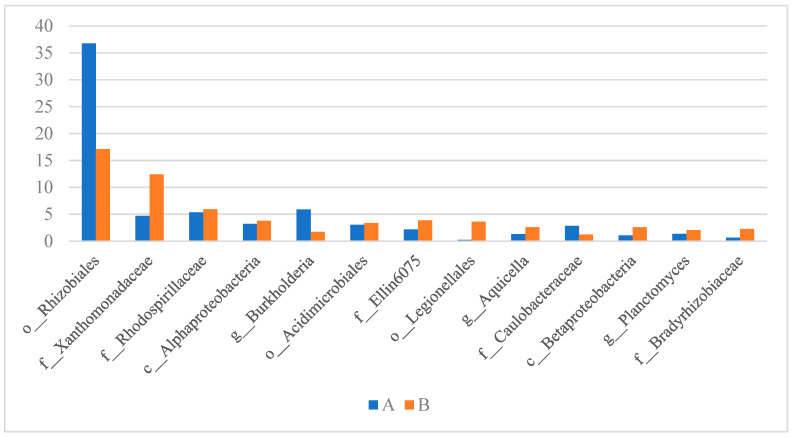
Detection rate of sequences at OTU level in samples A and B.

**Figure 8 materials-16-00619-f008:**
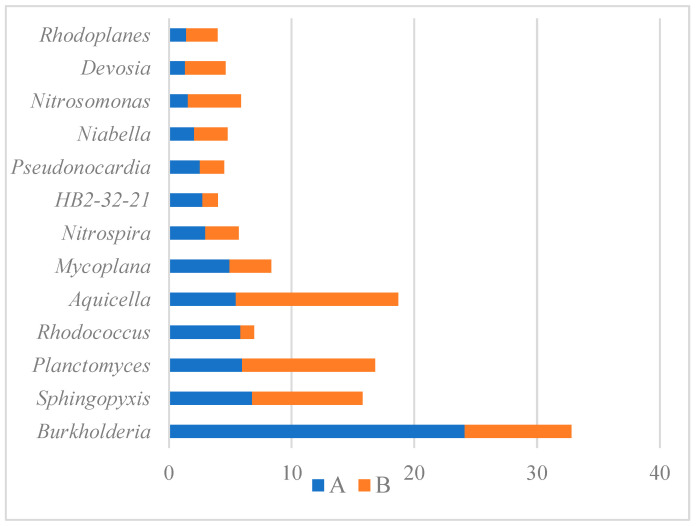
Percentage share of the most frequent identified genera in the examined samples.

**Figure 9 materials-16-00619-f009:**
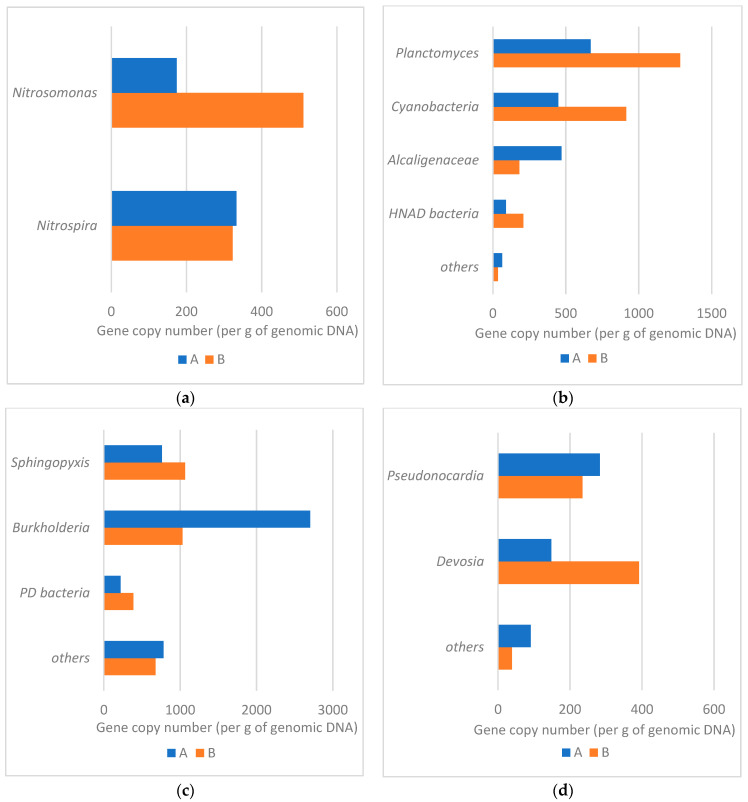
Bacteria participating in the process of (**a**) nitrification, (**b**) nitrification + denitrification, (**c**) denitrification, (**d**) phosphorus removal.

**Figure 10 materials-16-00619-f010:**
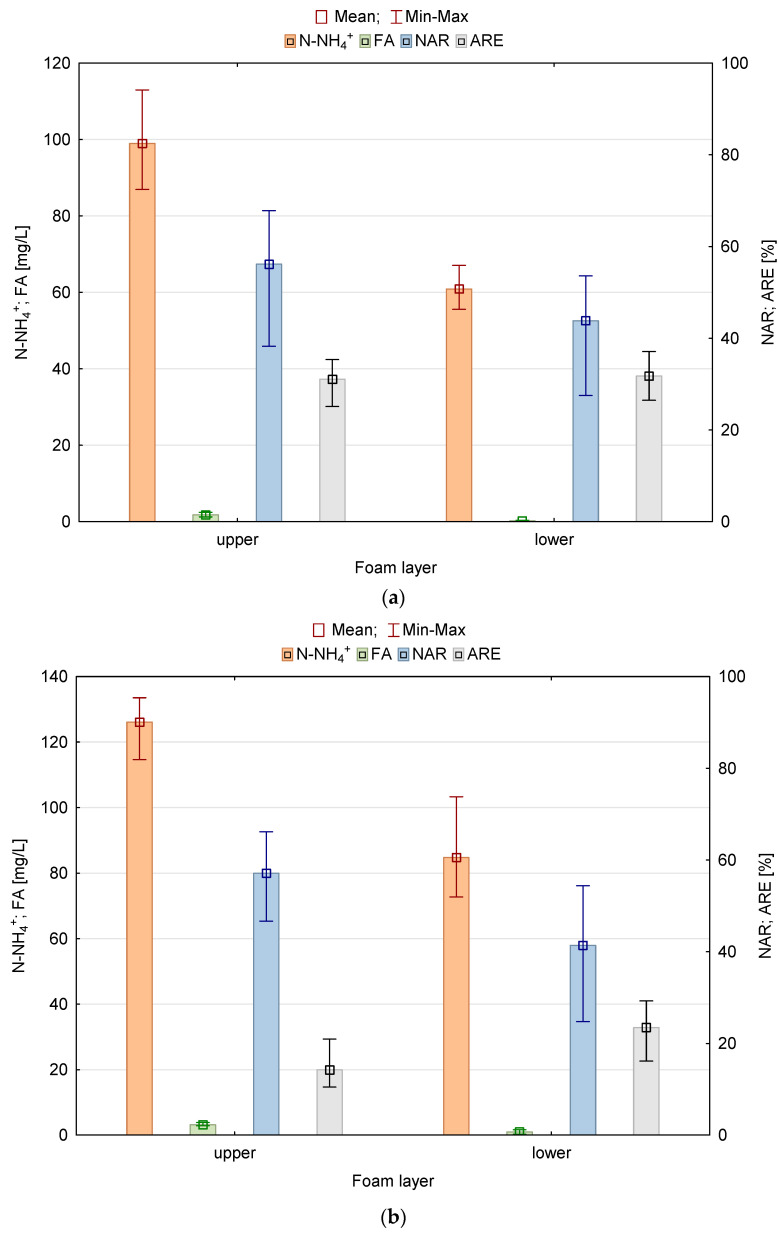
Box-plot chart for N-NH_4_^+^ concentration, FA, NAR, and reduction of ammonium nitrogen ARE in subsequent foam layers of (**a**) column A and (**b**) column B.

**Figure 11 materials-16-00619-f011:**
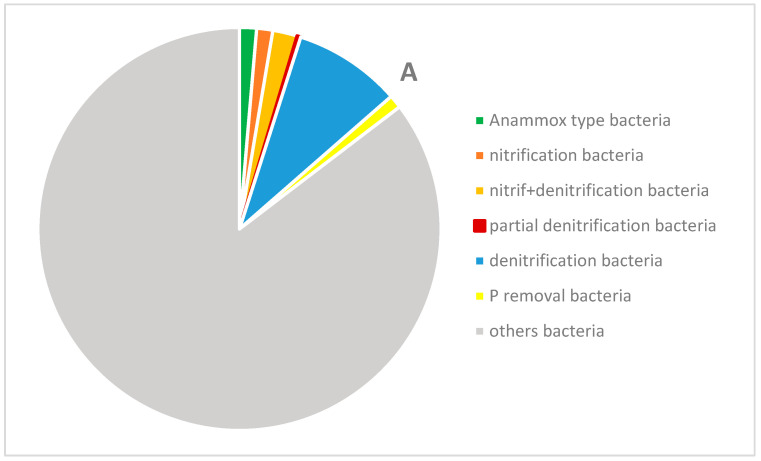
Bacterial community structure shown as the percentage share of anammox bacteria, nitrifiers, denitrifiers, and P-removing bacteria in the two types of PUR foam samples ((**A**) and (**B**) refer to the PUR foam samples (**A**) and (**B**), respectively).

**Table 1 materials-16-00619-t001:** Mean content of biomass immobilized on waste PUFs with open pores (group I) after 54 h of batch culture.

Amount of Biomass [g/1 g of Weighed Amount]			
Yellow-Green Foam	Bluish-Green Foam	Orange Foam	Lilac Foam
0.193	0.224	0.060	0.071

**Table 2 materials-16-00619-t002:** Average content of biomass immobilized on waste PUFs from groups I–III after 48 h of batch culture.

Groups	Type of Filling	Biomass [g/1 g of Weighed Sample]
I	soft bluish-green foam	0.2120
	soft yellow-green foam	0.1820
II	rigid white foam	0.0678
	rigid green foam	0.0636
III	pieces of conduit	0.0025
	conduit + soft bluish-green foam	0.7182

**Table 3 materials-16-00619-t003:** Diversity indices and microbial community structure at phylum level in the two types of PUR foam samples.

Groups	Samples	
	A	B
Total sequence reads	51,056	67,264
OTUs	828	872
Shannon diversity index	3.97	4.73
Simpson index	0.12	0.03
Margalef richness	46.12	45.03

## Data Availability

The data presented in this study are available on request from the corresponding author.

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
