# Peer review of "Waste Polyurethane Foams as Biomass Carriers in the Treatment Process of Domestic Sewage with Increased Ammonium Nitrogen Content"

_materials, 2023, doi:10.3390/ma16020619_

Round 1
Reviewer 1 Report
I appreciate the way the manuscript is drafted. It sounds so unique and novel. It would add more value if a few changes could be made before acceptance.

Author Response
The authors would like to thank the Reviewer for the insightful review and valuable comments.
Please see the attachment.

Reviewer 2 Report
This study focuses on microbial growth on waste polyurethane sponge materials for nitrogen and phosphorus removal from wastewater. It is an interesting topic. However, major revisions are still required before its final publication.
Specific comments
1. This manuscript should be well organized, in a better section and paragraph structure.
2. Abstract should be rewritten.
3. Introduction can be improved.
4. Please fix the format and grammar problems, for example: inconsistence of “Types of biomass carriers” in section 2.1.1 and “Types of biomass carriers” in section 2.2.1, case sensitive of “ACKNOWLEDGMENT” and “CONFLICTS OF INTEREST”.
5. According to the author guidance, check the following parts: Author Contributions; Funding; Institutional Review Board Statement; Informed Consent Statement; Data Availability Statement.
6. Summary and Conclusions: it is tediously long and should be improved.
7. The results need to be further summarized and refined, such as Figures 1~3 and Tables 1~2.
8. Figure 4 can be moved into the supplementary materials.
9. Figure 9 should be revised.
10. Figure 10 can be improved.
11. Further discussion on the microbial communities can be supplemented. Some relevant studies can be cited, Chemosphere, 2021, 278, 130436; Bioresource Technology, 2021, 324: 124668, i.e.
Author Response

(The authors gave the same response as above.)

Reviewer 3 Report
The topic of the work has relevance. The study's subject matter is correct and interesting.
It is worth noting that there are different font sizes in the abstract and the acknowledgments, which should be corrected.
For a better view, the microbial community composition (Table 3) should be presented as a circular diagram.
Figure 10 should be placed after the paragraph where it was first mentioned.
In conclusion, it is also worth outlining future research directions in the context of the continuation of the research topic.
Author Response

(The authors gave the same response as above.)

Round 2
Reviewer 2 Report
The authors have revised the manuscript carefully. I think the as-revised manuscript can be accepted for the publication.